# What Predicts Improvement of Dizziness after Multimodal and Interdisciplinary Day Care Treatment?

**DOI:** 10.3390/jcm11072005

**Published:** 2022-04-03

**Authors:** Tino Prell, Sigrid Finn, Hannah M. Zipprich, Hubertus Axer

**Affiliations:** 1Department of Geriatrics, Halle University Hospital, 06120 Halle, Germany; tino.prell@uk-halle.de; 2Center for Vertigo and Dizziness, Jena University Hospital, Friedrich Schiller University, 07743 Jena, Germany; sigrid.finn@med.uni-jena.de; 3Center for Healthy Ageing, Jena University Hospital, Friedrich Schiller University, 07743 Jena, Germany; hannah.zipprich@med.uni-jena.de; 4Department of Neurology, Jena University Hospital, Friedrich Schiller University, 07743 Jena, Germany

**Keywords:** chronic dizziness, vertigo, predictors, multimodal therapy

## Abstract

Background: Vertigo and dizziness are common in community-dwelling people and can be treated in specialized multidisciplinary settings. To develop tailored interventions, however, we have to explore risk factors for favorable and unfavorable outcomes. Methods: We prospectively investigated patients with chronic vertigo and dizziness subjected to our 5-day multimodal and interdisciplinary day care treatment in the Center for Vertigo and Dizziness of Jena University Hospital, Germany. The Vertigo Severity Scale (VSS), the Body Sensations Questionnaire (BSQ), the Hospital Anxiety and Depression Scale (HADS), the Agoraphobic Cognitions Questionnaire (ACQ), the Mobility Inventory (MI), and the burden and intensity of dizziness (using a visual analogue scale) were assessed at baseline (*n* = 754) and after 6 months (*n* = 444). In addition, 14 Likert-scaled questions were used to quantify the change in personal attitude and behavior towards the complaints after 6 months. Results: Dizziness-related burden and intensity improved with a large effect size. The largest improvement was seen in the attitudes towards dizziness, the understanding of somatic causes, and the perceived ability to influence dizziness. However, the ability to work and to carry out professional activity was improved to a lesser extent. The overall improvement of dizziness was associated with the absence of a depressive mood, a short duration of vertigo, a lower VSS, a lower perceived intensity of vertigo, and distinct vertigo diagnoses, namely Meniere’s disease, vestibular migraine, vestibular neuritis, vestibular paroxysmia, and vestibular schwannoma. Worsening of dizziness/vertigo was associated with depressive symptoms, permanent vertigo, distinct vertigo diagnoses (central vertigo, multisensory deficit), and a higher perceived burden due to vertigo. Conclusion: The six-month outcome of patients with dizziness presented to a specialized outpatient clinic appears to be favorable. Nevertheless, people with the abovementioned risk factors at baseline have less benefit and probably need adapted and tailored vertigo interventions to improve long-term outcome.

## 1. Introduction

Vertigo and dizziness are common medical complaints in everyday medical practice, with a lifetime prevalence of 7.4% in the general population aged 18–79 years [1,2]. These symptoms often impair quality of life and the ability to work. Moreover, they have a high risk for chronification [3] and complications (e.g., falls), especially in older adults [4]. Tailored diagnostic and therapeutic management are therefore urgently needed. However, it is also important to find out which patients benefit from specialized treatment, and which do not.

Dizziness or vertigo may be classified as chronic if symptoms have persisted at least for 3 months or attacks have recurred often in the last 3 months (≥5 days with symptoms/month) [5,6]. The chronification of symptoms makes therapy difficult and elaborate [7]. Therapeutic approaches for chronic dizziness/vertigo comprise pharmacotherapy [8,9], vestibular rehabilitation as an exercise-based therapy [10,11], and cognitive–behavioral therapy (CBT) [12]. In recent years, interdisciplinary multimodal therapy concepts have been developed [13,14] to address multifactorial influences for chronic dizziness/vertigo. One example of a specialized vertigo center is the Center for Vertigo and Dizziness of Jena University Hospital. It is a multidisciplinary outpatient clinic for patients with chronic vertigo, dizziness, or gait impairment. The interdisciplinary team consists of specialists in neurology, otolaryngology, and physical medicine and rehabilitation, as well as psychologists. Every patient subjected to the Center initially receives an individual diagnostic workup. The first evaluation is based on medical history, clinical examination, and the evaluation of findings from already performed technical diagnostics. Further technical diagnostics can be performed, such as advanced clinical neurophysiology and vestibular diagnostics, as well as imaging procedures such as CT or MRI—if considered to be necessary. In addition, every patient receives a psychological assessment. Consecutively, every patient receives a medical diagnosis, a therapeutic conception, and is evaluated if they may be suited for a 5-day multimodal day care treatment.

Such a thorough diagnostic process and specific treatment in a specialized care center exerts a long-term positive outcome [13]. To improve tailored interventions and outcome, it is essential to know the risk factors for a favorable and an unfavorable outcome. For instance, using the change in Dizziness Handicap Inventory (DHI) as an outcome measure, risk factors for an unfavorable outcome were advanced age, severe disability, constant vertigo or dizziness, and concomitant back pain [13]. Moreover, several studies investigated the outcome after specialized therapy in distinct disorders causing vertigo or dizziness [15,16,17]. However, data on the long-term outcome following diagnosis and the treatment of patients suffering from chronic neuro-otological disorders remain scarce. More data are needed to determine baseline predictors of subjective outcome after specialized therapy for chronic dizziness/vertigo. Here, we prospectively investigated all patients with chronic vertigo and dizziness subjected to our 5-day multimodal and interdisciplinary day care treatment.

## 2. Materials and Methods

### 2.1. Patients

We analyzed a database of patients who participated in our day care multimodal treatment program for chronic dizziness or vertigo. The data were prospectively collected between June 2013 and September 2017. Details about the cohort and assessments were described previously [5,18]. This study was carried out in accordance with the Declaration of Helsinki, and all participants gave written informed consent. The study was approved by the local ethics committee (number 5426-02/18). A total of 754 patients filled out a detailed questionnaire before therapy. Six months after attendance of the therapy week, the patients were contacted via mail and asked to fill out a questionnaire for follow-up assessment. Overall, 444 (58.9%) of the patients completed the questionnaire and sent it back. Thus, these questionnaires represent the follow-up status of the patients 6 months after attendance of the therapy week.

Multimodal and interdisciplinary day care treatment took place from Monday to Friday with an average of 7 h of therapy per day. The therapeutic team consisted of a nurse, a neurologist, a psychologist, and a physiotherapist. Elements of the multimodal group therapy were specific physiotherapeutic training, CBT-based psychoeducation and group therapy, training of Jacobson’s muscle relaxation technique, health education, specialized medical evaluation, and optimization of drug therapy [18]. Appendix A shows the time schedule of the therapy week. Group sizes varied between 8 and 10 patients. Every patient had an outpatient consultation with a neurologist, including a thorough diagnostic process, before the patient was subjected to the therapy week.

### 2.2. Assessments

Several parameters were assessed at baseline: age (metric; years), gender (nominal; male or female), duration of symptoms (nominal; 3–6 or >6 months), diagnosis category (nominal; somatic, non-somatic psychogenic, non-somatic unspecific), medical diagnosis as classified by the medical experts based on the International Classification of Vestibular Disorders (ICVD) of the Bárány Society, attacks of dizziness/vertigo (nominal; yes or no), and permanence of dizziness/vertigo (nominal; permanent present yes or no). In addition, the following patient-reported outcome measures were collected: the Vertigo Severity Scale (VSS), the Body Sensations Questionnaire (BSQ), the Hospital Anxiety and Depression Scale (HADS), the Agoraphobic Cognitions Questionnaire (ACQ), and the Mobility Inventory (MI). A visual analogue scale was used to measure the burden and intensity of dizziness (high values indicating severe dizziness on a scale from 0 to 10).

The Vertigo Severity Scale (VSS) allows for quantification of vertigo and dizziness symptoms [3], and the two subscales vestibular balance (VSS-V) and autonomic anxiety (VSS-A) can be calculated [19]. The 17 items of the BSQ measure anxiety about bodily sensations on a 5-point Likert scale [20], especially the BSQ1, which assesses the amount of fear. The ACQ is a 14-item questionnaire designed to measure maladaptive thoughts about the potential for catastrophic consequences arising from anxiety or panic [20]. The Mobility Inventory (MI) measures the degree to which places or situations are avoided with a trusted companion (MI-accompanied) and when the patient is alone (MI-alone) [21]. For the ACQ, the BCQ, and the MI, lower scores indicate better outcomes. The HADS is a frequently used self-rating scale developed to assess psychological distress. It consists of 7 items for the anxiety subscale (HADS anxiety) and 7 for the depression subscale (HADS depression). Each item is scored on a 0 to 3 response scale. After adjusting for six items that are reversed scored, all responses are summed to obtain the two subscales [22].

The patient-reported outcome measures (VSS, HADS, ACQ, BSQ, and MI) were selected to describe the cohort and to capture a representational quantification of vertigo and dizziness-related symptoms, i.e., anxiety- and depression-associated symptoms, maladaptive thoughts, avoidance behavior, and bodily sensations, as these play a central role in the therapy program.

At follow-up (6 months after attendance of the therapy week) all of these parameters were assessed a second time: the VSS, the HADS anxiety, the HADS depression, the ACQ, the BSQ, the MI, and the dizziness/vertigo-related burden and intensity of dizziness/vertigo via visual analogue scales as well.

In addition, 14 ordinal-scaled questions asked for changes of burden and perception of dizziness since treatment in the outpatient clinic 6 months prior. The questions were designed to estimate in what way the therapy week had influenced the personal attitude and behavior of the patients towards their complaints and were specifically tailored to the contents of our multimodal therapy program. The 14 questions comprised: (1) attitude towards vertigo/dizziness, (2) mental resilience, (3) physical resilience, (4) perceived ability to influence dizziness, (5) understanding of somatic causes, (6) understanding of psychological causes, (7) subjective quality of life, (8) general state of health, (9) performance of everyday activities (e.g., shopping, household, and hobbies), (10) performance of professional activities, (11) number of factors triggering dizziness/vertigo (e.g., movements/situations/stress), (12) intensity of unfavorable/negative thoughts about dizziness/vertigo, (13) dealing with external stressors (e.g., time pressure, many stimuli, and conflicts), and (14) dealing with internal stress-reinforcing attitudes (e.g., being perfect and wanting to do everything by oneself). These questions were rated on a Likert scale ranging from 0 to 5 (0 = worsening, 1 = no improvement at all, 2 = little improvement, 3 = moderate improvement, 4 = good improvement, and 5 = very good improvement). Cronbach’s alpha showed with 0.968 a high test score reliability.

### 2.3. Statistics

All data were analyzed with the Statistical Package for the Social Sciences software (version 25.0; IBM Corporation, Armonk, NY, USA), Jamovi (version 1.8.2.0; https://www.jamovi.org, (accessed on 9 February 2022), Sydney, Australia) or JASP (version 0.14.1.0; https://www.jasp-stats.org, (accessed on 9 February 2022), Amsterdam, The Netherlands). The values were presented as means and standard deviations (SD), and numbers and percentages. Normal distribution was determined using the Shapiro–Wilk test. First, we described the cohort using descriptive statistics. Changes in metric values between baseline and follow-up were compared using the Wilcoxon test. A principal component analysis was used to reduce the 14 questions concerning changes of burden and perception of dizziness to one compound factor. Linear regression with backward selection was used to determine the association between this compound factor and clinical variables (after exclusion of multicollinearity and autocorrelation). For all analyses, a *p* value < 0.05 was considered statistically significant.

## 3. Results

The baseline characteristics of the cohort are given in Table 1. Overall, 86.7% of the patients suffered from dizziness/vertigo for longer than 6 months and 13.3% between 3 and 6 months. It has to be noted that only patients with chronic dizziness/vertigo were subjected to the therapy program, which means that symptoms lasted for at least three months. The cut-off of 6 months was chosen from a pragmatic point of view as the persistence of symptoms over at least six months is a prerequisite for the diagnosis of a somatoform disorder.

There were no remarkable differences between people who completed or had not completed the follow-up (Appendix A), with the exception of statistically significant differences for the BSQ1 (*p* = 0.01), the VSS-A (*p* = 0.007), and the HADS depression (*p* = 0.019); however, the effect sizes of these differences were low.

The changes in vertigo-specific questionnaires from baseline to follow-up are given in Table 2. Here, we observed improvements for the BSQ1, the VSS, the HADS anxiety, and the MI alone. The largest effect size was found for the improvement of the VSS and its vestibular-balance (VSS-V) subscale. Less improvement was observed for the autonomic-anxiety (VSS-A) subscale.

In contrast, the HADS depression, the ACQ, and the MI-accompanied scales did not significantly differ between baseline and follow-up. Changes of dizziness-related burden were measured in two ways. First, a visual analogue scale was used to measure the burden and intensity at baseline and follow-up. For the mean, this VAS decreased by 1.4 ± 2.3 points for burden (from 6.41 ± 1.98 to 4.99 ± 2.15, *p* < 0.001, effect size by rank biserial correlation = 0.68), and 1.3 ± 2.2 points for intensity (from 5.94 ± 1.81 to 4.65 ± 1.80, *p* < 0.001, effect size by rank biserial correlation = 0.72) until follow-up (Figure 1). As indicated by the effect sizes, both improvements in the VAS can be regarded as strong effects. The distribution of the difference (improvement or worsening of dizziness) between the VAS scoring at baseline and follow-up are given in Figure 2.

For the second way to measure changes, 14 ordinal-scaled questions concerning changes of burden and the perception of dizziness were applied. Here, the majority reported different degrees of improvement. The largest improvement was seen in the attitudes towards dizziness, the understanding of somatic causes, and the perceived ability to influence dizziness (Figure 3; Appendix A). In contrast, the ability to work and to carry out professional activity was improved to a lesser extent (Figure 3). A principal component analysis revealed that the 14 questions concerning changes in attitude and behavior of the patients could be reduced to one factor: describing overall dizziness-related change from baseline to follow-up (Appendix A). Figure 4 shows the compound change factor for distinct diagnoses with the largest improvement for vestibular paroxysmia followed by vestibular Schwannoma and Meniere’s disease.

By using this compound factor, we then determined the baseline predictors for dizziness-related changes (Table 3). In the first model, we analyzed the predictive value of dizziness-related parameters. In the second model, we corrected for dizziness-related questionnaires (e.g., the BSQ, the ACQ, and the MI), and in the third model for age, gender, HADS anxiety, and HADS depression. The final model revealed that improvement of dizziness at follow-up was associated with the absence of a depressive mood, a short duration of vertigo, a lower VSS, and a lower perceived intensity of vertigo, and distinct vertigo diagnoses, namely Meniere’s disease, vestibular migraine, vestibular neuritis, vestibular paroxysmia, and vestibular schwannoma. Vice versa, a worsening of dizziness/vertigo was associated with depressive symptoms, permanent symptoms, distinct diagnoses (i.e., central vertigo and multisensory deficit), and a higher perceived burden due to dizziness/vertigo.

## 4. Discussion

We used two different ways to quantify the subjective outcome after a specialized multimodal day care treatment program. In general, we found that the six-month outcome of patients with chronic dizziness/vertigo appears to be favorable. Similar favorable effects of an outpatient therapy program have been shown by Obermann et al. [13] with a good and persistent outcome even after 2 years. However, not all patients might profit from a specialized vertigo therapy program, and we therefore aimed to determine baseline predictors for good and poor outcomes.

We identified five factors that contributed to an unfavorable outcome, i.e., the presence of depressive symptoms, a longer disease duration, permanent vertigo, a higher VSS, a higher perceived intensity of dizziness/vertigo, and distinct diagnoses (especially central vertigo and multisensory deficit). Vice versa, indicators for good improvements were the absence of a depressive mood, a short duration of symptoms, a lower VSS, and a lower perceived intensity of vertigo.

The relevance of perceived intensity and the initial VSS score for the outcome, agrees in principle with the findings from Obermann et al., who used the Dizziness Handicap Inventory (DHI) as an outcome measure after interdisciplinary treatment in a tertiary care neuro-otology institution [13]. In their study, risk factors for an unfavorable outcome were advanced age, severe disability, permanent vertigo or dizziness, and concomitant back pain, while depression and anxiety did not contribute considerably to the outcome [13]. This is quite plausible as it can be supposed that the more severely the dizziness-associated handicap is perceived and the longer the symptoms last, the more difficult and longer the therapy may be to reach an improvement.

The impact of the baseline extent of dizziness-associated handicap on follow-up was also found in other studies. In therapy-naive patients with chronic vestibular diseases, the baseline handicap explained that most of the variance correlated with the handicap at a 3-month follow-up [23]. In patients with functional vertigo and dizziness subjected to a multimodal psychosomatic inpatient treatment, the vertigo-related handicap at admission was the only significant predictor of vertigo-related handicap at follow-up [14]. Additionally, in patients with acute unilateral vestibular neuritis, the initial handicap was found to be associated with a higher handicap after vestibular rehabilitation [24]. Finally, initial disease severity was a predictor of the response to selective serotonin reuptake inhibitors in patients with persistent postural-perceptual dizziness (PPPD) [25].

We also found that a longer duration of symptoms was associated with an unfavorable outcome. This was also described in different conditions of dizziness/vertigo. In patients with benign paroxysmal positional vertigo (BPPV), a long duration of the disease was correlated with an increased rate of recurrence [26,27]. In addition, the persistence of residual dizziness after successful repositioning maneuvers in BPPV also depends on a longer symptom duration [28,29]. Patients with PPPD who did not benefit from vestibular rehabilitation therapy had a significantly longer duration of PPPD and a higher dizziness handicap inventory score than the patients who benefited [16].

Our study highlights the role of psychiatric comorbidities for the outcome. It is well known that vertigo and dizziness are associated with different psychiatric conditions [30]. The most common comorbidities are depression and anxiety [31,32,33].

Therefore, all of our patients were examined by a medical doctor and by a psychologist as well. In this cohort, a considerable amount showed anxiety (17% scored in HADS anxiety 8–10 and 16% ≥ 11) and depression (14% scored in HADS depression 8–10 and 12% ≥ 11). Of note, 7.6% showed symptoms of phobic postural vertigo, 13% symptoms of somatization, and 16.5% symptoms of secondary somatization.

It has been hypothesized that both psychiatric comorbidity and anxiety may be increased in people with vestibular excitation (such as vestibular migraine, vestibular paroxysmia, and Meniere’s disease), and be decreased in people with a loss of peripheral vestibular function (i.e., chronic unilateral and bilateral vestibulopathy) [34]. Meniere’s disease is especially susceptible to depression and anxiety [35,36,37,38], and PPPD as well [39]. On the other hand, patients with depressive disorders have an increased risk (1.55 times more likely) of developing BPPV [40]. Moreover, anxiety and/or depression symptoms significantly reduced the efficacy of the first-time repositioning maneuver in BPPV and increased the risk for a relapse [41].

We found that depression was particularly associated with an unfavorable outcome. It was demonstrated earlier that the rate of depression was increased in patients with audio-vestibular diseases in accordance with the presence of vertigo alone, vertigo accompanied by hearing loss, repeated symptoms, and bilateral hearing loss [42]. In patients with dizziness, associations between depression and anxiety with a self-rated severe disability could be demonstrated [43]. Dizzy patients with depression reported a significantly higher level of disability than dizzy patients without depression [43], and dizzy patients with psychiatric comorbidity reported more vertigo-related symptoms, and more depressive, anxiety, and somatization symptoms as compared to patients without a psychiatric comorbidity [31].

Vice versa, it was also described that dizziness as a somatic symptom significantly predicted major and minor depression in the follow-up [44]. An analysis of the longitudinal effects of vertigo and dizziness symptoms on anxiety and depression revealed that a lack of perceived control over symptoms may contribute to the development of depression [45]. Depression, anxiety, and somatization influenced vertigo symptoms and vertigo-related handicaps in a longitudinal observational study [46]. Therefore, we conclude that people with chronic dizziness and depression need additional and tailored interventions to improve outcome.

In addition, our data revealed better therapeutic effects in diagnoses mostly accompanied by attacks and excitation, i.e., Meniere’s disease, vestibular migraine, and vestibular paroxysmia, and in mainly a unilateral loss of function, i.e., unilateral vestibular neuritis and vestibular schwannoma. In contrast, worse therapeutic effects were achieved in mainly degenerative diseases with a greater loss of function, i.e., central vertigo and multisensory deficit.

Some limitations of our study have to be addressed. The patients were selected for participation in the therapy program depending on their diagnosis, motivation, dizziness/vertigo-associated disability, and physical and mental independence to participate in an outpatient program. Thus, patients with diagnoses such as BPPV were instead treated in a single session and self-reliant on repositioning maneuvers at home, while patients with diagnoses such as PPPD [18] may be over-represented. In addition, 58.9% of the patients filled out the follow-up questionnaires, which leads to selection bias.

## 5. Conclusions

In conclusion, the six-month outcome of patients with chronic dizziness/vertigo who participated in an interdisciplinary multimodal one-week outpatient therapy program appears to be favorable. Effect sizes were largest for improvements in the VSS-V and VSS total scores, which also represents clinically significant benefits for the patients. Nevertheless, people with depressive symptoms, longer disease duration, permanent dizziness/vertigo, higher severity, a higher perceived intensity of dizziness/vertigo, and distinct diagnoses (especially central vertigo and multisensory deficit) at baseline have less benefit and probably need adapted and tailored interventions to improve long-term outcome.

## Figures and Tables

**Figure 1 jcm-11-02005-f001:**
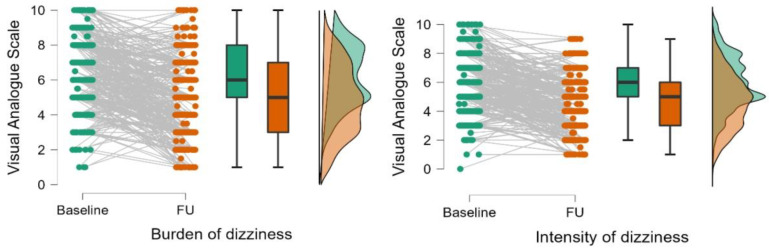
Burden and intensity of dizziness/vertigo (visual analogue scale) before and six months after therapy. Green color shows scores at baseline, orange color shows scores at follow-up (FU).

**Figure 2 jcm-11-02005-f002:**
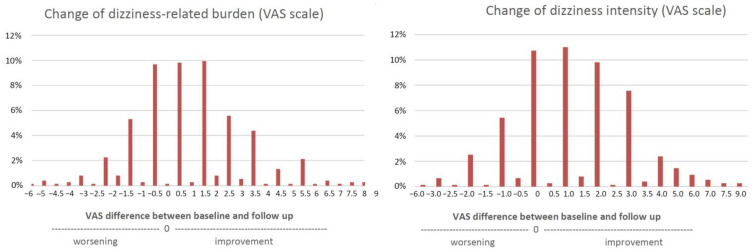
Change of burden and intensity of dizziness/vertigo.

**Figure 3 jcm-11-02005-f003:**
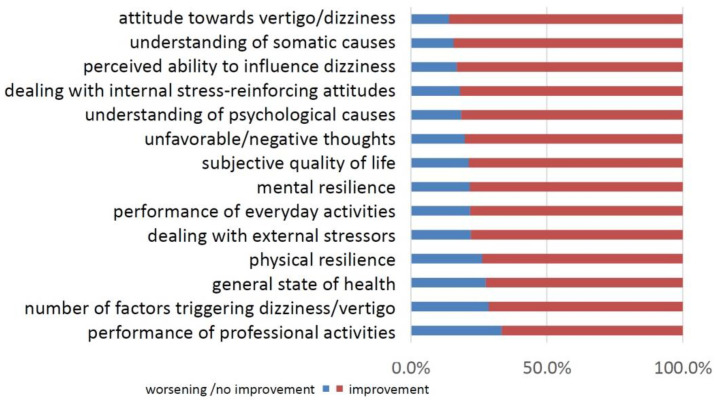
Changes in the last six months after therapy with categorization into improvement or no improvement at follow-up.

**Figure 4 jcm-11-02005-f004:**
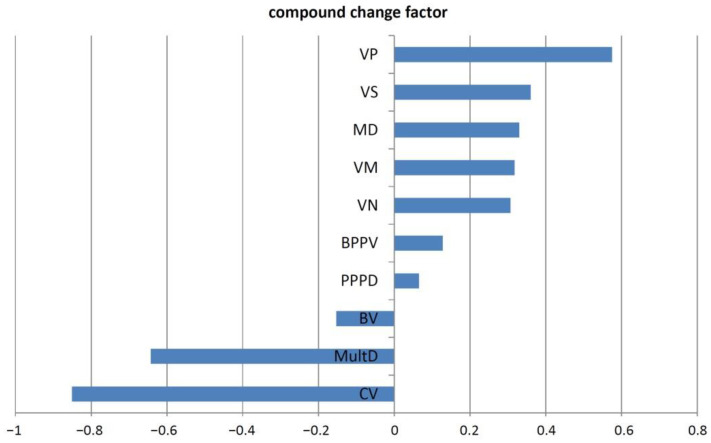
Compound change factor of diagnoses. A principal component analysis revealed that the 14 items concerning changes in attitude and behavior could be reduced to one compound factor. For each participant the regression value for this compound factor was calculated. Positive values on the *x*-axis indicate improvement and negative values indicate worsening of attitudes and behaviors towards dizziness at follow-up for distinct diagnoses. Abbreviations: BPPV, benign paroxysmal positional vertigo; BV, bilateral vestibulopathy; CV, central vertigo; MD, Meniere’s disease; MultD, multisensory deficit; PPPD, persistent postural-perceptual dizziness; VM, vestibular migraine; VN, vestibular neuritis; VP, vestibular paroxysmia; VS, vestibular schwannoma.

**Table 1 jcm-11-02005-t001:** Baseline characteristics of the cohort.

Categorical Variables			N	%
Sex	Female		461	61.5
	Male		289	38.5
Diagnosis category	Somatic		350	53.8
	non-somatic psychological		254	39.1
	Non-somatic unspecific		46	7.1
Duration of symptoms	3–6 months		95	13.3
	>6 months		621	86.7
Medical diagnosis	BPPV		24	3.2
	BV		35	4.6
	CV		43	5.7
	MD		55	7.3
	MultD		117	15.5
	PPPD		351	46.6
	VM		30	4.0
	VN		73	9.7
	VP		10	1.3
	VS		16	2.1
Continuous/permanent dizziness/vertigo	Yes		399	56.4
	No		308	43.6
Attack-like	Yes		402	60.5
	No		263	39.5
**Metric Variables**	**Mean**	**SD**	**95% Lower CI**	**95% Upper CI**
Age (years)	57.67	14.98	56.59	58.74
BSQ1	8.15	5.64	7.74	8.57
VSS-V	11.21	8.70	10.58	11.84
VSS-A	13.98	10.31	13.23	14.73
VSS total score	25.19	16.28	24.01	26.37
HADS anxiety	6.84	4.04	6.55	7.14
HADS depression	6.18	3.89	5.90	6.47
ACQ	19.63	5.84	19.13	20.14
MI accompanied	2.31	1.12	2.22	2.40
MI alone	1.92	0.96	1.84	2.01

Abbreviations: ACQ, Agoraphobic Cognitions Questionnaire; BPPV, benign paroxysmal positional vertigo; BSQ, Body Sensations Questionnaire; BV, bilateral vestibulopathy; CV, central vertigo; HADS, Hospital Anxiety and Depression Scale; MD, Meniere’s disease; MI, Mobility Inventory; MultD, multisensory deficit; PPPD, persistent postural-perceptual dizziness; VM, vestibular migraine; VN, vestibular neuritis; VP, vestibular paroxysmia; VS, vestibular schwannoma; VSS, Vertigo Severity Scale.

**Table 2 jcm-11-02005-t002:** Paired samples Wilcoxon signed-rank test: baseline vs. follow-up.

	Mean	SD	SE	*p*	Effect Size
BSQ1	7.751	5.178	0.266		
BSQ1 follow-up	6.860	5.754	0.296	<0.001	0.235
VSS-V	11.039	8.240	0.396		
VSS-V follow-up	8.477	8.424	0.405	<0.001	0.403
VSS-A	13.088	9.895	0.476		
VSS-A follow-up	11.963	9.320	0.448	0.008	0.151
VSS total score	24.127	15.427	0.742		
VSS total follow-up	20.440	15.688	0.755	<0.001	0.320
HADS-A	6.633	3.908	0.190		
HADS-A follow-up	6.111	4.002	0.195	<0.001	0.204
HADS-D	5.969	3.893	0.190		
HADS-D follow-up	5.668	3.779	0.184	0.110	0.098
ACQ	19.357	5.452	0.328		
ACQ follow-up	18.715	5.145	0.309	0.070	0.133
MI accompanied	1.925	0.952	0.059		
MI accompanied follow-up	1.977	0.914	0.056	0.182	−0.097
MI alone	2.317	1.054	0.065		
MI alone follow-up	1.647	0.812	0.050	<0.001	0.720

Abbreviations: ACQ, Agoraphobic Cognitions Questionnaire; BSQ, Body Sensations Questionnaire; HADS, Hospital Anxiety and Depression Scale; MI, Mobility Inventory; SD, standard deviation; SE, standard error; VSS, Vertigo Severity Scale.

**Table 3 jcm-11-02005-t003:** Linear regression: dependent variable: dizziness-related overall change/compound factor.

	Variable	Coefficient	SE	*p*	Beta
**Model 1 (corrected R^2^ = 0.13) ^1^**					
	Constant	0.173	0.107	0.106	
	Vertigo diagnoses (CV, MultD) #	−0.562	0.215	0.01	0.500
	Vertigo diagnoses (MD, VM, VN, VP, VS) #	0.317	0.138	0.022	0.500
	Permanent vertigo present	−0.369	0.124	0.003	0.295
	Disease duration < 6 m	0.453	0.182	0.014	0.205
**Model 2 (corrected R^2^ = 0.16) ^2^**					
	Constant	0.822	0.243	0.001	
	Vertigo diagnoses (CV, MultD) #	−0.452	0.215	0.036	0.306
	Vertigo diagnoses (MD, VM, VN, VP, VS) #	0.321	0.135	0.019	0.306
	Disease duration < 6 m	0.565	0.183	0.002	0.232
	Burden	−0.091	0.035	0.011	0.160
	Permanent vertigo present	−0.285	0.128	0.027	0.121
	MI alone	−0.142	0.064	0.027	0.120
	VSS	0.007	0.005	0.115	0.061
**Model 3 (corrected R^2^ = 0.30) ^3^**					
	Constant	1.070	0.220	<0.001	
	HADS depression	−0.102	0.015	<0.001	0.622
	Vertigo diagnoses (CV, MultD) #	−0.295	0.197	0.136	0.099
	Vertigo diagnoses (MD, VM, VN, VP, VS) #	0.246	0.124	0.049	0.099
	VSS	0.009	0.004	0.023	0.069
	Disease duration < 6 m	0.360	0.166	0.031	0.062
	Permanent vertigo present	−0.239	0.118	0.043	0.054
	Intensity	−0.067	0.034	0.049	0.052
	MI alone	−0.105	0.059	0.076	0.042

^1^ Model 1: Entered variables: permanent dizziness, attacks, disease duration, diagnosis category (psychogenic, somatic, unspecific), diagnosis. # Reference variable: diagnoses (BBPV, BV, PPPD). ^2^ Model 2: Entered variables: permanent dizziness, attacks, disease duration, diagnosis category, diagnosis, intensity of symptoms, burden due to dizziness/vertigo, BSQ1, VSS, ACQ, MI-accompanied, MI-alone # Reference variable: diagnoses (BPPV, BV, PPPD). ^3^ Model 3: permanent dizziness, attacks, disease duration, diagnosis category, diagnosis, intensity of symptoms, burden due to dizziness/vertigo, BSQ1, VSS, ACQ, MI-accompanied, MI-alone, BSQ1, age, gender, HADS anxiety, HADS depression. # Reference variable: diagnoses (BPPV, BV, PPPD). Abbreviations: ACQ, Agoraphobic Cognitions Questionnaire; BPPV, benign paroxysmal positional vertigo; BSQ, Body Sensations Questionnaire; BV, bilateral vestibulopathy; CV, central vertigo; HADS, Hospital Anxiety and Depression Scale; MD, Meniere’s disease; MI, Mobility Inventory; MultD, multisensory deficit; PPPD, persistent postural-perceptual dizziness; SE, standard error; VM, vestibular migraine; VN, vestibular neuritis; VP, vestibular paroxysmia; VS, vestibular schwannoma; VSS, Vertigo Severity Scale.

## Data Availability

The data used to support the findings of this study are available from the corresponding author upon request.

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
