# Peer review of "What Predicts Improvement of Dizziness after Multimodal and Interdisciplinary Day Care Treatment?"

_jcm, 2022, doi:10.3390/jcm11072005_

Round 1

Reviewer 1 Report

Dear Authors,

Thank you for your manuscript ID jcm-1614189.

It is known that balance disorders require a multidisciplinary diagnostic and rehabilitation approach, taking into great consideration the patient's follow-up and psychological attitude. These concepts are well highlighted in your study. Figures, Tables and References are pertinent.

In fact, I have only a few comments, in order to add value to your study and will appreciate the point-by-point response if possible.

1) Please specify all abbreviations the first time you use them. See PMR-line53; CBT 88 line.

2) In lines 81-84 it is reported that only 58.9% of the enrolled patients completed the questionnaire. As a great part of the sample did not complete the 6-months follow-up (41.1%), it is mandatory to specify the reasons.

3) Specify the median follow-up period in the sample (also specify range).

4) Patient reported outcome measures collected in your series were VSS, BSQ, HADS, ACQ, and MI. What are the criteria you considered in this selection? I think it is appropriate to specify it (why, for example, not consider the "Fall Risk Questionnaire", or "Balance Tinetti Scale", or other else?).

5) In the discussion, you stressed the importance of the patient's psychological and psychiatric attitude and comorbidities for the outcome. What is the percentage of patients with depression, anxiety and somatization in your series? Is the diagnosis obtained through a specialist evaluation?

Thank you so much for your work.

Best regards.

Author Response

Reviewer 1

COMMENT: It is known that balance disorders require a multidisciplinary diagnostic and rehabilitation approach, taking into great consideration the patient's follow-up and psychological attitude. These concepts are well highlighted in your study. Figures, Tables and References are pertinent. In fact, I have only a few comments, in order to add value to your study and will appreciate the point-by-point response if possible.

ANSWER: We thank the reviewer for this insightful feedback and comment.

COMMENT: 1) Please specify all abbreviations the first time you use them. See PMR-line53; CBT 88 line.

ANSWER: We are thankful to the Reviewer for pointing out these omissions on our part and revised accordingly throughout the manuscript (e.g. lines 47, 52).

COMMENT: 2) In lines 81-84 it is reported that only 58.9% of the enrolled patients completed the questionnaire. As a great part of the sample did not complete the 6-months follow-up (41.1%), it is mandatory to specify the reasons.

ANSWER: We specified this point in the text: ‘Six months after attendance of the therapy week the patients were contacted via mail and asked to fill out a questionnaire for follow-up assessment. Here, 444 (58.9%) of the patients completed the questionnaire and sent it back.’ (lines 82-86) As in many surveys there is a plethora of reasons for not taking part in the follow-up. Since, we did not contact the patients a second time, the reasons for not sending the questionnaire back are unfortunately not known.

COMMENT: 3) Specify the median follow-up period in the sample (also specify range).

ANSWER: We are appreciative of  this advice.  Questionnaires for follow up were sent to the patients 6 months after attendance of the therapy week. We added in the revised version: ‘Thus, these questionnaires represent the follow-up status of the patients 6 months after attendance of the therapy week.’ (lines 85-86)

COMMENT: 4) Patient reported outcome measures collected in your series were VSS, BSQ, HADS, ACQ, and MI. What are the criteria you considered in this selection? I think it is appropriate to specify it (why, for example, not consider the "Fall Risk Questionnaire", or "Balance Tinetti Scale", or other else?).

ANSWER: We are thankful for this relevant comment. We clarified this in the revised version: ‘The patient reported outcome measures collected in the questionnaires (VSS, HADS, ACQ, BSQ, and MI) were chosen because we intended to collect a representational quantification of vertigo and dizziness related symptoms, anxiety and depression associated symptoms, maladaptive thoughts, avoidance behavior and bodily sensations as these play a central role in the therapy program.’ (lines 122-126) We agree that the Fall Risk Questionnaire, for instance, would have been of additional value, but we were restricted to a pragmatic amount of work to fill out the questionnaires.

COMMENT: 5) In the discussion, you stressed the importance of the patient's psychological and psychiatric attitude and comorbidities for the outcome. What is the percentage of patients with depression, anxiety and somatization in your series? Is the diagnosis obtained through a specialist evaluation?

ANSWER: We detailed these aspects in the revision: ‘Therefore, all of our patients were examined by a medical doctor and by a psychologist as well. In this cohort a considerable amount showed anxiety (17% scored in HADS anxiety 8-10 and 16% ≥11) and depression (14% scored in HADS depression 8-10 and 12% ≥11). Of note, 7.6% showed symptoms of phobic postural vertigo, 13% symptoms of somatization and 16.5% symptoms of secondary somatization.’ (lines 294-298).

COMMENT: Thank you so much for your work.

ANSWER: Thank you very much. We appreciate the suggestions and hope that we have done this to the Reviewer’s satisfaction in the revised version.

Reviewer 2 Report

This is an important study on a topic of great concern to seniors and with potential severe consequences after falls. It is meticulously conducted, analysed and presented. English is perfect (however, readers outside Europe will not know what anamnesis is).

Methods:

Please describe your interesting 5 day intervention in detail. Readers will want to know and perhaps emulate your protocol.

“Vertigo Severity Scale (VSS), Body Sensations Questionnaire (BSQ), Hospital Anxiety and Depression Scale (HADS), Agoraphobic Cognitions Questionnaire (ACQ), and Mobility Inventory (MI) were assessed at baseline (n=754) and after 6 months (n=444). The burden and intensity of dizziness (using a visual analogue scale) and 14 Likert-scaled questions were used as outcome measures.”

[please explain why you chose two of these measures as the outcomes and not also the others]

[please explain derivation of the 14 item Likert scale and any validity and reliability measures available]

Thus is very important.

Please explain why you created a nominal outcome for duration of symptoms (< 6 and > 6 months). Please present data on how much information you are missing by so doing  How many had symptom resolution or improvement at shorter times?

Results.

Your prospective cohort began with 754 patients and at 6 months 444 were evaluable. You commented that at 6 months there were no "remarkable differences" between those with and without 6 month follow up. However, in Table S1 the differences for BSQ1 are p = .01, VSS-A .007 and HADS depression .019. 

Could you please perform an ITT (intention to treat) analysis to convince readers that the results for the 444 are the same as for the 754?  

Conclusions

Please comment whether you found that the statistically significant differences in BSQ1, VSS-V, VSS-A, UVSS-Total and HADS-A are clinically significant for the patients?

Author Response

Reviewer 2

This is an important study on a topic of great concern to seniors and with potential severe consequences after falls. It is meticulously conducted, analysed and presented. English is perfect (however, readers outside Europe will not know what anamnesis is).

ANSWER: Thank you. We changed ‘anamnesis’ into ‘medical history’. (Line 54)

COMMENT: Methods: Please describe your interesting 5 day intervention in detail. Readers will want to know and perhaps emulate your protocol.

ANSWER: We thank the Reviewer for making this relevant suggestion. We added a table in the supplement (Supplementary Table S1) with a detailed time schedule and added to the text: ‘Supplementary Table S1 shows the time schedule of the therapy week. Group sizes varied between 8 and 10 patients. Every patient had an outpatient consultation with a neurologist including a thorough diagnostic process before the patient was subjected to the therapy week.’ (lines 92-95)

COMMENT:  “Vertigo Severity Scale (VSS), Body Sensations Questionnaire (BSQ), Hospital Anxiety and Depression Scale (HADS), Agoraphobic Cognitions Questionnaire (ACQ), and Mobility Inventory (MI) were assessed at baseline (n=754) and after 6 months (n=444). The burden and intensity of dizziness (using a visual analogue scale) and 14 Likert-scaled questions were used as outcome measures.”  [please explain why you chose two of these measures as the outcomes and not also the others]

ANSWER: Kindly excuse the lack of clarity in the first version. Our primary endpoint in the study was dizziness-related burden/intensity assessed by the 14 ordinal rated questions and two visual analogue scales. The other measures were used to describe the cohort and used as independent variables to determine which parameters influence the primary outcome measures.  We rephrased in the abstract accordingly: ‘In addition, 14 Likert-scaled questions were used to quantify the change of personal attitude and behavior towards the complaints.’ (lines 18-20)  And in the methods section:  ‘At follow-up (6 months after attendance of the therapy week) all of these parameters were assessed a second time:  VSS, HADS anxiety, HADS depression, ACQ, BSQ, MI, and the dizziness/vertigo related burden and intensity of dizziness/vertigo via visual analogue scales as well.’ (lines 127-130)

COMMENT:  [please explain derivation of the 14 item Likert scale and any validity and reliability measures available]

ANSWER: Thank you for this comment. We clarified this issue: ‘The questions were designed to estimate in what way the therapy week has influenced the personal attitude and behavior of the patients towards their complaints and were specifically tailored to the contents of our multimodal therapy program.’ (lines 132-136) As a measure of reliability we provided the Cronbach’s alpha which was 0.968, indicating excellent reliability (line 147). Criterion-related validity was not determined, as there is no existing, validated tool intended to measure the same construct. Nevertheless, we have some data indicating that the 14 items (and the corresponding compound score) are valid. We also asked participants to rate on a visual analogue scale how dizziness changed after treatment. Here, the VAS rating for “change of subjective burden in everyday life due to dizziness” (r=0.75, p < 0.001) and “change of attention to the symptom of dizziness in everyday life” (r = 0.63, p < 0.001) correlated with the compound score, indicating construct validity.

COMMENT: Please explain why you created a nominal outcome for duration of symptoms (< 6 and > 6 months). Please present data on how much information you are missing by so doing  How many had symptom resolution or improvement at shorter times?

ANSWER: We specified it as follows: ‘86.7% of the patients suffered from dizziness/vertigo longer than 6 months and 13.3% between 3 and 6 months.  It has to be noted that only patients with chronic dizziness/vertigo were subjected to the therapy program which means that symptoms lasted at least three months. The cut-off of 6 months was chosen from a pragmatic point of view as the persistence of symptoms over at least six months is a prerequisite for the diagnosis of a somatoform disorder.’ (lines 169-175)

COMMENT: Results. Your prospective cohort began with 754 patients and at 6 months 444 were evaluable. You commented that at 6 months there were no "remarkable differences" between those with and without 6 month follow up. However, in Table S1 the differences for BSQ1 are p = .01, VSS-A .007 and HADS depression .019.

ANSWER: We thank the reviewer for pointing this out. We used the term “no remarkable difference”, because the significant differences only had a small effect size. Significance or p-value per se should not be used as indicator for a meaningful difference. However, we agree that this is misleading and revised accordingly: ‘There were no remarkable differences between people who completed or not completed follow-up (Supplementary Table S2) with the exception of statistical significant differences for BSQ1 (p=0.01), VSS-A (p=0.007) and HADS depression (p=0.019); however the effect sizes of these differences were low.´ (lines 176-179)

COMMENT: Could you please perform an ITT (intention to treat) analysis to convince readers that the results for the 444 are the same as for the 754? 

ANSWER: We thank the reviewer for this question. Although it is uncommon, it is true that ITT can also be used in observational studies. However, as the missing data exceed 30% and there is only one follow-up, one should not use imputation methods to overcome the problem of missing data. Therefore, from a statistical point of view a ITT is not possible with the existing dataset.

COMMENT: Conclusions: Please comment whether you found that the statistically significant differences in BSQ1, VSS-V, VSS-A, UVSS-Total and HADS-A are clinically significant for the patients?

ANSWER: We thank the reviewer for this question. We used the effect sizes to determine if significant differences were relevant. We included this statement in the conclusions: ‘Effect sizes were largest for improvements in VSS-V and VSS total score, which represents also clinically significant benefits for the patients.’ (lines 342-343)

Reviewer 3 Report

The paper reads well with proper use of English. The manuscript length is appropriate.

The abstract is adequately clear to reveal the nature of the article.

The presentation of the rationale of the work was logical and clear.

In the introduction, lines 54- 57, the authors stated that:The first evaluation is based on anamnesis, clinical examination, and available diagnostic findings. Further technical diagnostics can be performed such as advanced clinical neurophysiology, vestibular diagnostics as well as imaging procedures such as CT or MRI – if considered to be necessary”.

“Available diagnostic findings”. Do they mean provisional diagnosis provided from the history and clinical examination alone (key symptoms and/or signs?) It would better if they could provide some detail or examples of such “available diagnostic findings”.

In the methodology, lines 133-134, the authors stated that:ranging from 0 to 5 (0=worsening, 1=no improvement at all, 2=little improvement, 3=moderate improvement, 4=fair improvement, and 5=strong improvement)”.

Was a fair improvement considered a more improvement than moderate improvement?

In the results, lines 189-190, the authors stated that:Figure 4 shows the compound change factor for distinct diagnoses with largest improvement for disorders like vestibular paroxysmia and Meniere´s disease”.

However, figure 4 shows that the largest improvement was for vestibular paroxysmia then vestibular Schwannoma and thirdly, Meniere´s disease”.

In the results, lines 186-188 and 194-196, the authors stated that:

“A principal component analysis revealed that the 14 questions concerning changes in attitude and behavior of the patients could be reduced to one factor, describing overall dizziness-related change from baseline to follow-up (Supplementary Table S3)”. “By using this compound factor, we then determined the baseline predictors for dizziness-related changes (Table 3)”.

Data analyses are appropriate  but this statistical analyses in the results section are somewhat complex for the non-statistician reader to understand. Supplementary Table S3 and figure 4 need some clarification for the non-statistician reader.

In Supplementary Table S2, the 4th row: " Attention to dizziness/vertigo”, this item was not included in the 14 ordinal rated questions, nor in the intensity of dizziness/vertigo VAS or the burden due to dizziness/vertigo VAS,

Was that an additional VAS assessment not mentioned in the methodology? Please clarify.

Overall, the work reported in the paper is interesting and of clinical application.

Author Response

Reviewer 3

COMMENT: The paper reads well with proper use of English. The manuscript length is appropriate. The abstract is adequately clear to reveal the nature of the article. The presentation of the rationale of the work was logical and clear.

ANSWER: We thank the reviewer for this encouraging comment.

COMMENT: In the introduction, lines 54- 57, the authors stated that: “The first evaluation is based on anamnesis, clinical examination, and available diagnostic findings. Further technical diagnostics can be performed such as advanced clinical neurophysiology, vestibular diagnostics as well as imaging procedures such as CT or MRI – if considered to be necessary”. “Available diagnostic findings”. Do they mean provisional diagnosis provided from the history and clinical examination alone (key symptoms and/or signs?) It would better if they could provide some detail or examples of such “available diagnostic findings”.

ANSWER: We thank the reviewer for pointing this out. We revised for clarity: ‘The first evaluation is based on medical history, clinical examination, and the evaluation of findings from already performed technical diagnostics.’ (lines 54-56)

COMMENT: In the methodology, lines 133-134, the authors stated that: “ranging from 0 to 5 (0=worsening, 1=no improvement at all, 2=little improvement, 3=moderate improvement, 4=fair improvement, and 5=strong improvement)”. Was a fair improvement considered a more improvement than moderate improvement?

ANSWER: Sorry, the translation from German was mistakable. We changed the descriptions as follows: ‘0=worsening, 1=no improvement at all, 2=little improvement, 3=moderate improvement, 4=good improvement, and 5=very good improvement’ (lines 145-147). Supplementary Figure S1 was changed accordingly.

COMMENT: In the results, lines 189-190, the authors stated that: “Figure 4 shows the compound change factor for distinct diagnoses with largest improvement for disorders like vestibular paroxysmia and Meniere´s disease”. However, figure 4 shows that the largest improvement was for vestibular paroxysmia then vestibular Schwannoma and thirdly, Meniere´s disease”.

ANSWER: We thank the reviewer for this advice. We revised accordingly (lines 213-215).

COMMENT: In the results, lines 186-188 and 194-196, the authors stated that: “A principal component analysis revealed that the 14 questions concerning changes in attitude and behavior of the patients could be reduced to one factor, describing overall dizziness-related change from baseline to follow-up (Supplementary Table S3)”. “By using this compound factor, we then determined the baseline predictors for dizziness-related changes (Table 3)”. Data analyses are appropriate  but this statistical analyses in the results section are somewhat complex for the non-statistician reader to understand. Supplementary Table S3 and figure 4 need some clarification for the non-statistician reader.

ANSWER: We thank the reviewer for this suggestion. We added description of PCA in the corresponding table and figure. (Figure 4, Supplementary Table S4)

COMMENT: In Supplementary Table S2, the 4th row: " Attention to dizziness/vertigo”, this item was not included in the 14 ordinal rated questions, nor in the intensity of dizziness/vertigo VAS or the burden due to dizziness/vertigo VAS. Was that an additional VAS assessment not mentioned in the methodology? Please clarify.

ANSWER: We thank the reviewer for pointing this out. This was a mistake, as this question was not part of the current analyses. Therefore, it was deleted in the revised version. (Supplementary Table S3)

COMMENT: Overall, the work reported in the paper is interesting and of clinical application.

ANSWER: We than the reviewer for the valuable comments and advices.

Round 2

Reviewer 2 Report

Thanks to the authors for their revisions. An excellent, carefully planned and analysed study.